# Targeted *L*_1_-Regularization and Joint Modeling of Neural Networks for Causal Inference

**DOI:** 10.3390/e24091290

**Published:** 2022-09-13

**Authors:** Mehdi Rostami, Olli Saarela

**Affiliations:** Dalla Lana School of Public Health, University of Toronto, Toronto, ON M5T 3M7, Canada

**Keywords:** causal Inference, instrumental variables, neural networks, doubly robust estimation, regularization

## Abstract

The calculation of the Augmented Inverse Probability Weighting (AIPW) estimator of the Average Treatment Effect (ATE) is carried out in two steps, where in the first step, the treatment and outcome are modeled, and in the second step, the predictions are inserted into the AIPW estimator. The model misspecification in the first step has led researchers to utilize Machine Learning algorithms instead of parametric algorithms. However, the existence of strong confounders and/or Instrumental Variables (IVs) can lead the complex ML algorithms to provide perfect predictions for the treatment model which can violate the positivity assumption and elevate the variance of AIPW estimators. Thus the complexity of ML algorithms must be controlled to avoid perfect predictions for the treatment model while still learning the relationship between the confounders and the treatment and outcome. We use two NN architectures with an $L_{1}$-regularization on specific NN parameters and investigate how their certain hyperparameters should be tuned in the presence of confounders and IVs to achieve a low bias-variance tradeoff for ATE estimators such as AIPW estimator. Through simulation results, we will provide recommendations as to how NNs can be employed for ATE estimation.

## 1. Introduction

There are generally two approaches to address causal inference in observational studies. The first one is to draw population-level causal inference which goes back at least to the 1970s [1]. The second is to draw conditional causal inference which has received attention more recently [2,3]. An example of a population-level causal parameter the average treatment effect (ATE),
(1)$$\beta_{ATE}=E\left[Y^{1}-Y^{0}\right]=E\left[E[Y^{1}-Y^{0}|W]\right].$$

The quantity $E[Y^{1}-Y^{0}|W]$ is referred to as the conditional average treatment effect (CATE) [4,5,6,7,8,9,10]. CATE is NOT an individual-level causal parameter. The latter is impossible to estimate accurately unless both potential outcomes are observed for each individual (under parallel worlds!), or *W* contains all the varying factors that make the causal relationship deterministic, which are unlikely to hold in practice. That said, under certain assumptions, the counterfactual loss, the loss due to the absence of counterfactual outcome, can be upper bounded [11]. The present article focuses on the estimation of ATE which does not require those assumptions.

Through a number of attempts, researchers have utilized ML models for the causal parameter estimation [12,13,14,15,16,17]. While the ultimate goal of a ML algorithm is to predict the outcome of interest as accurately as possible, it does not optimally serve the main purpose of the causal parameter estimation. In fact, ML algorithms minimize some prediction loss containing the treatment or the observed outcome (and not counterfactual outcome) and without targeting any relevant predictor(s) such as confounding variables [18].

Including confounders for the estimation of ATE in observational studies avoids potential selection bias [19], however, in practice, we do not have a priori knowledge about the confounders and the ML algorithm minimizes the loss function without discriminating between the input covariates. In fact, the ML algorithm can successfully learn the linear and non-linear relationships between the confounders and the treatment and outcome, but at the same time, might learn from potential Instrumental Variables (IVs) present in the data as well (the variables that predict the treatment, but not the outcome). If there are strong confounders or IVs among the covariates, the predictions of treatments (i.e., the propensity scores) can become extreme (near zero or one) which in turn can make the estimates unstable. While possibly reducing the bias, the variance gets elevated at the same time. Less complex models, on the other hand, may suffer from large bias (under-fitting) but can obtain more stable causal parameter estimation. This conflict between the necessary complexity in the model(s) and the bias-variance tradeoff motivates to develop ML algorithms for step 1 that provide a compromise between learning from confounders and IVs to entail a balance between the bias and variance of the causal parameter in step 2. In addition to a low bias-variance tradeoff, the asymptotic normality of the causal effect estimator is wanted for inferential statistics.

Chernozhukov et al. [16] investigated the asymptotic normality of orthogonal estimators of ATE (including Augmented Inverse Probability Weighting (AIPW)) when two separate ML algorithms model the treatment and outcome, referred to as the Double Machine Learning (DML). With the same objective, Farrell et al. [17] utilized two separate neural networks (we refer to as the double NN or dNN), without the usage of any regularization other than using the Stochastic Gradient Descent (SGD) for model optimization. SGD does impose some regularization but is insufficient to control the complexity of NN algorithms where strong predictors exist in the data [20]. Rostami and Saarela [20] experimentally showed that when AIPW is utilized, dNN performs poorly. The normalization of AIPW helps control both the bias and variance of the estimator. Further, they illustrated that imposing the $L_{1}$ regularization on all of the parameters (without targeting a specific set of input features) helps reduce the bias, variance, and Mean Square Error (MSE) of the ATE estimators up to some extent. Simulations indicated that when dNN is used, with or without regularization, the normalized AIPW (nAIPW) outperforms AIPW. For a comprehensive literature review on the doubly robust estimators (including AIPW) see Moosavi et al. [21].

The strategy of targeting a specific type of features can be designed in NN architectures along with the necessary optimization and regularization techniques. Flexible NN structures, optimizations and regularization techniques are easily programmed in deep learning platforms such as pytorch.

Shi et al. [22] proposes a neural network architecture, referred to as the DragonNet, that jointly models the treatment and outcome, in which a multi-tasking optimization technique is employed. In the DragonNet architecture, the interaction of the treatment and non-linear transformations of the input variables are considered. Chernozhukov et al. [23] uses the Riesz Representer [16] as the minimizer of a stochastic loss, which provides an alternative for the propensity score estimation, and aims to prevent the empirical consistency assumption violation issue [20]. Chernozhukov et al. [23] also use the joint modeling of the Riesz Representer and the outcome through multi-tasking, and they call their method auto Double Machine Learning (Auto-DML). Chernozhukov et al. [24] optimized an $L_{1}$ regularized loss function to estimate weights rather than estimating propensity scores and plugging them into the AIPW estimator. Chernozhukov et al. [25] proposed optimizing a minimax loss function for the same purpose. In this body of work, it is still unclear how to hyperparameter tune the chosen NN architecture for causal inference, especially for the ATE estimation.

Other techniques of feature selection before propensity score estimation have been proposed in the literature [26]. However, hard thresholding might ignore important information hidden in the features.

The objective of this research is to experimentally investigate how NN-type methods can be utilized for ATE estimation, and how the hyperparameters can be tuned to achieve the best bias-variance tradeoff for the ATE estimators. This is done in the presence of strong IVs and confounders. The papers cited above do not consider this general scenario.

In this research our goal is not any of the following: 1. We do not aim to compare NNs with other ML algorithms to see which ones outperform the others. By the no-free-lunch theorem, [27], there is no specific algorithm that can learn all relationships sufficiently well. Thus, it is expected that some ML algorithms are better in some scenarios and other algorithms in other scenarios. 2. We do not aim to study different types of causal parameters. 3. We do not aim to study different estimators of the Average Treatment Effect. 4. We do not aim to study feature selection or other types of methods that can prevent IVs to feed into the model of the treatment in the first step inference.

Throughout this research, we utilize nAIPW as it outperforms AIPW estimator in the presence of strong confounders and IVs [20]. To target the relevant inputs, we propose two methods. First, employing a type of $L_{1}$ regularization on top of the common $L_{1}$ regularization on all the network parameters. Second, we propose a joint model of the treatment and outcome in a Neural Network (jNN) architecture where we place both the treatment and outcome on the output layer of a multi-layer perceptron [28]. This NN architecture is appealing as it models the treatment and outcome simultaneously which can potentially target the relevant covariates that are predictive of both treatment and outcome (or confounder) and can mitigate or ignore the IVs’ effects on the predictions. We will investigate if both or either of these ideas improves the bias-variance tradeoff of the causal effect estimator as compared to a dNN model.

In this research, the NN architecture that jointly models the treatment and outcome here referred to as jNN. The parameters or weights are estimated by minimizing a regularized multi-task loss which is the summation of the Cross-Entropy (for modeling the binary treatment) and MSE loss (for modeling the continuous outcome) [29]. Multi-task learning can help target the predictors of both treatment and outcome that are placed in the output layer, and also it helps to resist against over-fitting in case of many irrelevant inputs [30]. Other benefits of multi-task learning are listed in Section 2.2. Also, two types of $L_{1}$ regularization terms are used in order to dampen the instrumental variables and strong confounders.

To show the effectiveness of jNN and dNN, a thorough simulation study is performed and these methods are compared in terms of the number of confounders and IVs that are captured in each scenario, the prediction measures, and the bias and variance of causal estimators. To investigate whether our network targets confounders rather than IVs and also dampens the impact of strong confounders on the propensity scores, we calculate the bias-variance tradeoff of causal estimators (i.e., minimal MSE) utilizing the NN predictions; Low bias means the model has mildly learned from confounders and other types of covariates for the outcome, and low variance means the model has ignored IVs and has dampened strong confounders in the treatment model. Further, a comparison between the methods is made on the Canadian Community Health Survey (CCHS) dataset where the intervention/treatment is food security versus food insecurity and the outcome is individuals’ body mass index (BMI).

The organization of this paper is as follows. In Section 1.2 we define the problem setting and the causal parameter to be estimated. In Section 2 we introduce the NN-type methods, their loss functions, and hyperparameters. Section 3 provides a quick review of the ATE estimators. In Section 4 our simulation scenarios are stated along with their results in Section 4.2. The results of the application of our methods on a real dataset are presented in Section 5. We conclude the paper in Section 6 with some discussion on the results and future work.

### 1.1. Notation

Let data $O=(O_{1},O_{2},...,O_{n})$ be generated by a data generating process *F*, where $O_{i}$ is a finite dimensional vector $O_{i}=\left(Y_{i},A_{i},W_{i}\right)$, with *Y* as the outcome, *A* as the treatment and $W=(X_{c},X_{y},X_{iv},X_{irr})$, where we assumes $A=f_{1}\left(X_{c},X_{iv}\right)+\epsilon_{1}$, and $Y=f_{2}\left(A,X_{c},X_{y}\right)+\epsilon_{2}$, for some functions $f_{1},f_{2}$. $X_{c}$ is the set of confounders, $X_{iv}$ is the set of instrumental variables, $X_{y}$ is the set of y-predictors (independent of the treatment), and $X_{irr}$ is a set of given noise or irrelevant inputs (Figure 1). *P* is the true observed data distribution, ${\hat{P}}_{n}$ is the distribution of O such that its marginal distribution with respect to *W* is its empirical distribution, and the expectation of the conditional distribution Y|A=a,W, for a=0,1, can be estimated. We denote the prediction function of observed outcome given covariates in the treated group $q^{1}:=q\left(1,W\right)=E\left[Y|A=1,W\right]$, and that in the untreated group $q^{0}:=q\left(0,W\right)=E\left[Y|A=0,W\right]$, and the propensity score as g(W)=E[A|W]. Throughout, the expectations E are with respect to *P*. The symbol ^ on the population-level quantities indicates the corresponding finite sample estimator, and *P* is replaced by ${\hat{P}}_{n}$.

### 1.2. Problem Setup and Assumptions

The fundamental problem of causal inference states that individual-level causality cannot be exactly determined since each person can experience only one value of *A*. Thus, it is customary to only estimate a population-level causal parameter, in this research Average Treatment Effect (ATE) (Equation 1).

For identifiablity of the parameter, the following assumptions must hold true. The first assumption is the Conditional Independence, Ignorability or Unconfoundedness stating that, given the confoudners, the potential outcomes are independent of the treatment assignments ($Y^{0},Y^{1}⊥A|W$). The second assumption is Positivity which entails that the assignment of treatment groups is not deterministic (0<Pr(A=1|W)<1) ([18], page 344). The third assumption is Consistency which states that the observed outcomes equal their corresponding potential outcomes ($Y^{A}=y$). There are other modeling assumptions made such as time order (i.e., the covariates *W* are measured before the treatment), IID subjects, and a linear causal effect.

## 2. Prediction Models

Neural Networks (NNs) are complex nonparametric models that approximate the underlying relationship between inputs and the outcome. The objective in causal inference, however, is not necessarily to leverage the maximum prediction strength of NNs and in fact, the NN architecture should be designed and tuned so that it pays more attention to the confounders.

The most important requirement of ML models such as NNs in causal inference is that although the outcome prediction model should minimize the corresponding loss (fit to get the best outcome prediction possible), given all of the covariates, the loss function associated with the propensity score model should not necessarily be minimized. Ideally, the instrumental variables or strong confounders which can give extreme fitted probability values (near zero or one) should be controlled when minimizing the loss. This can help prevent the elevation of the variance of the causal estimator (i.e., prevent the violation/near violation of the positivity assumption [18,31]). In summary, the prediction models should be strong enough to learn the linear and non-linear relationships between the confounders and treatment, but should not provide perfect predictions. We hypothesize that the employed NNs methods with the regularization techniques have the property of ignoring or damping strong confouders and/or instrumental variables.

### 2.1. Joint Neural Network

The joint Neural Network (jNN) architecture is a combination of multiple ideas (see Section 2.2, Section 2.3 and Section 2.4) for causal parameter estimation purposes mentioned above.

The jNN models are:(2)$$\left[\begin{array}{c} E[Y|A,W]\\ E[A|W] \end{array}\right]=\left[\begin{array}{c} \alpha_{0}+\beta A+W\alpha +H\Gamma_{Y}\\ g(\gamma_{0}+W\gamma +H\Gamma_{A}) \end{array}\right]$$
where $H=f(f\left(...\left(f\left(W\Omega_{1}\right)\Omega_{2}\right)...\right)\Omega_{L})$ is the last hidden layer matrix which is a non-linear representation of the inputs (*L* is the number of hidden layers), *g* is the logistic link function, and $\Gamma_{A}$ and $\Gamma_{Y}$ are the parameters that regress H to the log-odds of the treatment assignment or to the outcome in the output layer. The large square brackets around the equations above is meant to emphasize that both treatment and outcome models are trained jointly. The non-linear relationships between the inputs and the treatment and outcome can have arbitrary forms (which might not be the same for the treatment and outcome). The NNs can approximate such non-linear relationships even though one activation function is used. In fact, this property of NNs frees us from pre-specifying basis functions [26] as they can be estimated automatically.

The jNN architecture minimizes a multi-task loss Section 2.2 to estimate the networks parameters:(3)$$\begin{array}{cc} L\left(P,\beta ,\alpha \right)= & a\sum_{i=1}^{n}{\left[Y_{i}-\alpha^{\prime }-\beta A_{i}-W_{i}\alpha -H_{i}^{T}\Gamma_{Y}\right]}^{2}+\\ & b\sum_{i=1}^{n}\left[A_{i}\log\left(g\left(H_{i}^{T}\Gamma_{A}\right)\right)+\left(1-A_{i}\right)\log\left(1-g\left(H_{i}^{T}\Gamma_{A}\right)\right)\right]+\\ & \;\;\;\;\;\;\;\;\;\;C_{L_{1}}\sum_{\omega \in P}\left|\omega \right|+C_{L_{1TG}}\left(\sum_{\omega \in \Gamma_{A}}\left|\omega \right|+\sum_{\omega \in \Omega_{1}}\left|\omega \right|\right), \end{array}$$
where $a,b,C_{L_{1}},C_{L_{1TG}}$ are hyperparameters, that can be set before training or be determined by Cross-Validation, that can convey the training to pay more attention to one part of the output layer.

The jNN can have an arbitrary number of hidden layers, or the width of the network (H) is another hyperparameter. For a 3-layer network, $H=[l_{1},l_{2},...,l_{h}]$, where $l_{j}$ is the number neurons in layer *j*, j=1,2,...,h. $P=\{\omega \in \Omega_{1}\cup \Omega_{2}\cup \Omega_{3}\cup \Gamma_{Y}\cup \Gamma_{A}\}$, are the connection parameters in the nonlinear part of the network, with Ω’s being shared for the two outcome and propensity models. Noted that the number of parameters with $L_{1}$ regularization (third term on (Equation 3)) is $\left|P\right|=\left(p+1\right)\times l_{1}+\left(l_{1}+1\right)\times l_{2}+...+\left(l_{h-1}+1\right)\times l_{h}+\left(l_{h}+1\right)\times 2$, including the intercepts in each layer.

The following subsections list the potential benefits and the rationale behind the proposed network (Equations (Equation 2) and (Equation 3)).

### 2.2. Bivariate Prediction, Parameters Sharing, and Multi-Task Learning

One of the main components of the jNN architecture is that both treatment and outcome are placed and modeled in the output layer simultaneously. The hypothesis here is that the network learns to get information from the inputs that predict both treatment and outcome, i.e., the confounders. This bivariate structure is intertwined with a multi-task learning or optimization. Ruder [30] reviews the multi-tasking in machine learning and lists its benefits such as implicit data augmentation, regularization, attention focusing, Eavesdropping and Representation bias. Caruana [32] showed that overfitting declines by adding more nodes to the output layer as compared to modeling each output separately Baxter [33]. The multi-task is used when more than one output is used. Multi-task learning is common in the field of Artificial Intelligence and Computer Vision, for example, for the object detection task where the neural network predicts the coordinates of the box around objects and also classifies the object(s) inside the box (see for example [34]). Multi-task learning is used in jNN in order to investigate if the model pays more attention to the confounders than other types of inputs.

### 2.3. Regularization

The jNN will be resistant to overfitting by adding regularization to the network. Preliminary simulations revealed that $L_{2}$, and the Dropout Goodfellow et al. [35] regularization techniques do not result in satisfactory causal effect estimation, and the inherent regularization in the Stochastic Gradient Descent Goodfellow et al. [35] is also insufficient, while $L_{1}$ regularization is effective. We did not use the early-stopping as a regularization technique.

The $L_{1}$ regularization, third summation in (Equation 3), shrinks the magnitude of the parameter estimates of the non-linear part of the architecture which, in effect, limits the influence of $X_{irr}$ and $X_{iv}$, $X_{y}$, and $X_{c}$ on both treatment and the outcome. The motivation behind the $L_{1}$ regularization is to avoid overfitting for better generalization.

The ideal situation for causal parameter estimation is to damp the instrumental variables and learn from confounders and y-predictors only. Henceforth another version of the $L_{1}$ regularization is introduced here, referred to as the targeted $L_{1}$ regularization, or $L_{1TG}$, to potentially reduce the impact of instrumental variables on the outcome and more importantly on the propensity scores. The motivation is that by introducing shrinkage on the connections between the last hidden layer and the treatment, the neural network is trained to learn more about confounders than IVs in the last hidden layer as the outcome model is free to learn as much as possible from confounders. The caveat here might be that if the last hidden layer is large enough, some of the neurons can learn confounders while other learn from IVs, thus motivating to consider limiting the number of neurons in the last hidden layer. These hypotheses and ideas are considered in the simulation studies.

### 2.4. Linear Effects and Skip Connections

The terms βA+Wα and Wγ in (Equation 2) are responsible for potential linear effects. Theoretically, the non-linear parts of the NNs can estimate linear effects, but it is preferable to use linear terms if the relationship between the some of the inputs and the outcome/treatment are linear for more accurate linear effect estimation. The benefit of including linear terms in the equations has been verified in our preliminary simulation studies.

These linear terms are referred to the skip-connections in ML literature He et al. [36] which connect some layers to two or more layers forward. In ML literature, they are primarily used in very deep neural networks to facilitate optimizations. But they are used in jNN to model the linear effects directly. More specifically, skip connections connect the covariates to both treatment and outcome in the output layers and connect the treatment in the input layer to the outcome in the output layer. The latter skip connection is shown in Figure 2. It should be noted that this skip connection in particular is independent of the treatment in the output layer to avoid perfect prediction of the propensity scores.

### 2.5. Double Neural Networks

In order to study the significance of the proposed method through simulations, we compare jNN with the double Neural Networks (dNN) Chernozhukov et al. [37] method. dNN is generally referred to the strategy of modeling the treatment and outcome separately utilizing two different models:(4)$$\begin{array}{cc} E[Y|A,W] & =\beta_{0}+\beta A+W\alpha +H\Gamma_{Y}\\ E[A|W] & =\alpha_{0}+U\alpha +K\Gamma_{A}, \end{array}$$
where two separate neural nets model *y* and *A* (no parameter sharing). In this paper, the dNN algorithm refers to two neural networks to model the treatment and outcome separately. To make the two jNN and dNN models comparable, we let the NN architectures to be as similar as possible in terms of skip connections and regularization techniques. The loss functions in dNN to be optimized are:(5)$$\begin{array}{cc} \; & L_{y}\left(P_{y},\beta ,\alpha \right)=\sum_{i=1}^{n}{\left[Y_{i}-\alpha^{\prime }-\beta a_{i}-W_{i}\alpha -H_{i}^{T}\Gamma_{Y}\right]}^{2}+C_{L_{1}}^{\prime }\sum_{\omega \in P}\left|\omega \right|,\\ \; & L_{A}\left(P_{A}\right)=\sum_{i=1}^{n}\left[a_{i}\log\left(g\left(K_{i}^{T}\Gamma_{A}\right)\right)+\left(1-a_{i}\right)\log\left(1-g\left(K_{i}^{T}\Gamma_{A}\right)\right)\right]+C_{L_{1}}^{\prime \prime }\sum_{\omega \in P}\left|\omega \right|+\\ & C_{L_{1TG}}\left(\sum_{\omega \in \Gamma_{A}}\left|\omega \right|+\sum_{\omega \in \Omega_{1}}\left|\omega \right|\right), \end{array}$$

## 3. ATE Estimation

The Causal Parameter Estimation algorithm is a two stage process. The regression functions E[A|W],E[Y|A=1,W]),E[Y|A=0,W] are estimated using the ML algorithms such as jNN or dNN in step 1. And in step 2, the predictions are inserted into the causal estimators such as (Equation 6), below.

### ATE Estimators

There is a wealth of literature on how to estimate the ATE and there are various versions of estimators including the Augmented Inverse Probability Weighting (AIPW), Normalized Augmented Inverse Probability Weighting (nAIPW):(6)$$\begin{array}{cc} {\hat{\beta }}_{AIPW} & =\frac{1}{n}\sum_{i=1}^{n}\left(\frac{A_{i}\left(Y_{i}-{\hat{q}}_{i}^{1}\right)}{{\hat{g}}_{i}}-\frac{\left(1-A_{i}\right)\left(Y_{i}-{\hat{q}}_{i}^{0}\right)}{1-{\hat{g}}_{i}}\right)+\frac{1}{n}\sum_{i=1}^{n}{\hat{q}}_{i}^{1}-{\hat{q}}_{i}^{0},\\ {\hat{\beta }}_{nAIPW} & =\sum_{i=1}^{n}\left(\frac{A_{i}\left(Y_{i}-{\hat{q}}_{i}^{1}\right)w_{i}^{\left(1\right)}}{\sum_{j=1}^{n}A_{j}w_{j}^{\left(1\right)}}-\frac{\left(1-A_{i}\right)\left(Y_{i}-{\hat{q}}_{i}^{0}\right)w_{i}^{\left(0\right)}}{\sum_{j=1}^{n}\left(1-A_{j}\right)w_{j}^{\left(0\right)}}\right)+\frac{1}{n}\sum_{i=1}^{n}{\hat{q}}_{i}^{1}-{\hat{q}}_{i}^{0}. \end{array}$$
where ${\hat{q}}_{i}^{k}=\hat{q}\left(k,W_{i}\right)=\hat{E}\left[Y_{i}|A_{i}=k,W_{i}\right]$ and ${\hat{g}}_{i}=\hat{E}\left[A_{i}|W_{i}\right]$, and $A_{1}$ is the treatment group with size $n_{1}$ and $A_{0}$ is the treatment group with size $n_{1}$.

In the second step of estimation procedure, the predictions of the treatment (i.e., propensity score, PS) and/or the outcome $\hat{E}\left[Y_{i}|A_{i}=k,W_{i}\right]$, k=0,1, can be inserted in these estimators (Equation 6). Generalized Linear Models (GLM), any relevant Machine Learning algorithm such as tree-based algorithms and their ensemble Friedman et al. [28], SuperLearner Van der Laan et al. [38], or Neural Network-based models (such as ours) can be applied as prediction models for the first step prediction task. We will use jNN and dNN in this article.

## 4. Simulations

A simulation study (with 100 iterations) was performed to compare the prediction methods jNN, and dNN by inserting their predictions in the nAIPW (causal) estimators (Equation 6). There are a total of 8 scenarios according to the size of the data (i.e., the number of subjects and number of covariates), and the confounding and instrumental variables strengths. We fixed the sample sizes to be n=750 and n=7500, with the number of covariates p=32 and p=300, respectively. The four sets of covariates had the same sizes $\#X_{c}=\#X_{iv}=\#X_{y}=\#X_{irr}=8.75$ and independent from each other were drawn from the Multivariate Normal (MVN) Distribution as X∼N(0,Σ), with $\Sigma_{kj}=\rho^{j-k}$ and ρ=0.5. Let β=1. The models to generate the treatment assignment and outcome were specified as
(7)$$\begin{array}{cc} A & ∼Ber\left(\frac{1}{1+e^{-\eta }}\right),\mathrm{with}\;\eta =f_{a}\left(X_{c}\right)\gamma_{c}+g_{a}\left(X_{iv}\right)\gamma_{iv},\\ Y & =3+A+f_{y}\left(X_{c}\right)\gamma_{c}^{\prime }+g_{y}\left(X_{y}\right)\gamma_{y}+\epsilon , \end{array}$$

The functions $f_{a},g_{a},f_{y},g_{y}$ select 30% of the columns and apply interactions and non-linear functions listed below (Equation 8). The strength of instrumental variable and confounding effects were chosen as $\gamma_{c},\gamma_{c}^{\prime },\gamma_{y}∼Unif\left(r_{1},r_{2}\right)$ where $\left(r_{1}=r_{2}=0.1\right)$ or $\left(r_{1}=0.1,r_{2}=1\right)$, and $\gamma_{iv}∼Unif\left(r_{3},r_{4}\right)$ where $\left(r_{3}=r_{4}=0.1\right)$ or $\left(r_{3}=0.1,r_{4}=1\right)$.

The non-linearities for each pair of covariates are randomly selected among the following functions:(8)$$\begin{array}{cc} \; & l\left(x_{1},x_{2}\right)=e^{\frac{x_{1}x_{2}}{2}}\\ \; & l\left(x_{1},x_{2}\right)=\frac{x_{1}}{1+e^{x_{2}}}\\ \; & l\left(x_{1},x_{2}\right)={\left(\frac{x_{1}x_{2}}{10}+2\right)}^{3}\\ \; & l\left(x_{1},x_{2}\right)={\left(x_{1}+x_{2}+3\right)}^{2}\\ \; & l\left(x_{1},x_{2}\right)=g\left(x_{1}\right)\times h\left(x_{2}\right) \end{array}$$
where g(x)=−2I(x≤−1)−I(−1≤x≤0)+I(0≤x≤2)+3I(x≥2), and h(x)=−5I(x≤0)−2I(0≤x≤1)+3I(x≥1), or g(x)=I(x≥0), and h(x)=I(x≥1).

In order to find the best set of hyperparameter values for the NN architectures, we ran an initial series of simulations to find the best set of hyperparameters for all scenarios, presented here. The networks’ activation function is Rectified Linear Unit (ReLU), with 3 hidden layers as large as the input size (p), with $L_{1}$ regularization and batch size equal to 3∗p and 200 epochs. The Adaptive Moment Estimation (Adam) optimizer Kingma and Ba [39] with learning rate 0.01 and momentum 0.95 were used to estimate the network’s parameters, including the causal parameter (ATE).

As in practice the RMSE and covariate types are unknown, prediction measures of the outcome and treatment should be used to choose the best model in a K-fold cross-validation. $R^{2}$ and AUC each provide insight about the outcome and treatment models, respectively, but in our framework, both models should be satisfactory. To measure the goodness of the prediction models (jNN and dNN) for causal inference purposes, we define and utilize a statistic which is a compromise (geometric average) between $R^{2}$ and AUC, here referred to as geo,
(9)$$geo\left(R,D\right)=\sqrt[{3}]{R^{2}\times D\times \left(1-D\right)},$$
where D=2(AUC−0.5), the Somers’ D index. This measure was not utilized in the optimization process (i.e., training the neural networks), and is rather introduced here to observe if the compromise between $R^{2}$ and AUC agrees with the models that capture more confounders than IVs. We will refer to geo(R,D) simply as geo.

### 4.1. Selected Covariate Types

In order to identify which types of covariates (confounders, IVs, y-predictors, and irrelevant covariates) the prediction methods have learned from, we calculate the association between the inputs and the predicted values ($\hat{E}\left[Y|A,W\right]$ and $\hat{E}\left[A|W\right]$), and after sorting the inputs (from large to small values) based on the association values, we count the number of different types of covariates within top 15 inputs. The association between two variables here is estimated using the distance correlation statistic [40] whose zero values entail independence and non-zero values entail statistical dependence between the two variables.

### 4.2. Results

Figure 3, Figure 4, Figure 5, Figure 6, Figure 7 and Figure 8 present the overall comparison of different hyperparameter settings of jNN and dNN architectures in terms of five different measures, respectively: (1) The average number of captured confounders/IVs/y-predictors, (2) Average Root Mean Square Error (RMSE) of causal estimators, (3) Average $R^{2}$, AUC and their mixture measure geo (Equation 9), (4) Bias, (5) MC standard deviation of nAIPW. The bootstrap confidence intervals for the bias, standard deviation and RMSE are calculated to capture significant differences between the simulation scenarios. The *x*-axis includes 16 hyperparameter settings, and as a general rule here, models in the left are most complex (less regularization and wider neural nets) and in the right are least complex. Noted that $L_{1TG}$ regularization is only targeted at the treatment model.

The Figure 3 and Figure 4 show how the complexity of both dNN and jNN (*x*-axis) impact the number of captured covariate types (i.e., confounders/IVs/y-predictors) (top graph), RMSE (middle graph) and prediction measures (bottom graph). In almost all the hyperparameter settings, especially when $C_{L_{1TG}}$ is non zero, the number of captured confounders is larger and the number of captured IVs is smaller in jNN as compared to dNN. This shows the joint modeling has a benefit of focusing on the confounders, rather than IVs, especially in the large data scenario.

The RMSE of jNN is larger than that of dNN for models with zero targetted regularization (the scenarios in the left). With decreasing the complexity of the treatment model, the RMSE of both jNN and dNN decline. The jNN outperforms dNN in almost all of the hyperparameter settings in case of n=750, but does not show a clear pattern in case of n=7500. Further, the impact of the width of architectures (*H*) changes based on $C_{L_{1}}$ regularization: wider architectures (H=[p,p,p], *p*: number of covariates) with large $C_{L_{1}}$ outperform other combinations of these two hyperparameters. This observation is more clear for smaller sized data, and for dNN model. In the small size scenarios, when the width is small (H=[3,32,3]), the outcome model is affected and has a smaller $R^{2}$. This means there are not enough neurons (on the first or last layer) to provide more accurate outcome predictions. In the best scenarios, the RMSE confidence intervals of jNN model are below those of dNN, illustrating a small preference of jNN over dNN in terms of RMSE. Comparing the three hyperparameters, $C_{L_{1TG}}$ is most effective, and zero values of this hyperparameter results in very large RMSEs for both dNN and jNN.

From Figure 3 and Figure 4, it is observed that both jNN and dNN models have roughly the same values for the $R^{2}$ (outcome model performance) across hyperparameter settings and for both data sizes (n=7500, and n=750). That is, the targeted regularization in jNN does not impact the performance of the outcome model. The AUC, on the other hand, declines with higher values of $C_{L_{1TG}}$, and is almost always smaller or equal in dNN as compared to jNN. Further, larger values of geo in the small size data correspond to smaller RMSE, but no such pattern can be seen in the large data scenario.

Overall, the trends favor the idea that more complex treatment models capture larger number of IVs, have larger AUC (smaller geo.), and have larger RMSE. That is, more complex models are less favorable.

Figure 7 and Figure 8 illustrate the bias and standard deviation of the causal estimators. As expected and mentioned in the Section 1, the models that do not dampen IVs suffer from large bias and standard deviation. The bias and standard deviation have opposite behavior in different settings, such that settings that produce larger standard deviation, results in small bias, and vice versa, except for the one setting that produces both largest bias and standard deviation. The fluctuations of the bias-variance across hyperparameter settings are larger in n=750 case than in n=7500 case. For small sample n=750, the best scenario for jNN is $H=\left[32,32,32\right],C_{L_{1}}=0.1,C_{L_{1TG}}=0.7$ where both bias and standard deviation of jNN are small in the same direction. For the large sample n=7500, however, the best scenario for jNN is $H=\left[30,300,30\right],C_{L_{1}}=0.01,C_{L_{1TG}}=0.7$ with a similar behavior. The best scenarios for dNN are slightly different. For small sample $H=\left[32,32,32\right],C_{L_{1}}=0.1,C_{L_{1TG}}=0.7$ and for the large sample $H=\left[30,300,30\right],C_{L_{1}}=0.01,C_{L_{1TG}}=0.7$ are most favorable.

## 5. Application: Food Insecurity and BMI

The Canadian Community Health Survey (CCHS) is a cross-sectional survey that collects data related to health status, health care utilization and health determinants for the Canadian population in multiple cycles. The 2021 CCHS covers the population 12 years of age and over living in the ten provinces and the three territorial capitals. Excluded from the survey’s coverage are: Persons living on reserves and other Aboriginal settlements in the provinces and some other sub-populations that altogether represent less than 3% of the Canadian population aged 12 and over. Examples of modules asked in most cycles are: General health, chronic conditions, smoking and alcohol use. For the 2021 cycle, thematic content on food security, home care, sedentary behavior and depression, among many others, was included. In addition to the health component of the survey are questions about respondent characteristics such as labor market activities, income and socio-demographics.

In this article, we use the CCHS dataset to investigate the causal relationship of food insecurity and body mass index (BMI). Other gathered information in the CCHS is used which might contain potential confounders, y-predictors and instrumental variables. The data are from a survey and need special methods such as the resampling or bootstrap methods to estimate the standard errors. However, here, we use the data to illustrate the utilization of jNN and dNN with different hyperparameters choices in the presence of possible empirical positivity violations. In order to reduce the amount of variability in the data, we have focused on the sub-population 18–65 years of age.

Figure 5 and Figure 6 present the ATE estimates and their 95% asymptotic confidence intervals with nIPW, AIPW and nAIPW methods. Figure 5 contains hyperparameter settings where there is no targeted regularization and it shows how important this regularization technique is, especially for the AIPW estimator that has no normalization. We have removed these scenarios in Figure 6 for a more clear comparison between the remaining scenarios. The estimates and 95% CIs seem similar across the hyperparameter settings, but there is a clear difference between those of AIPW and nAIPW. This means that for this dataset, normalization might not be needed as the propensity scores do not behave extremely and AIPW does not blow up.

## 6. Discussion

In this paper, we have studied how hyperparameters of the Neural Network predictions in the first step can affect the Average Treatment Effect (ATE) estimator. We have considered a general Data Generating Process (DGP) that four types of covariates that exist in the dataset, confounders, IVs, y-predictors, and irrelevant covariates. Two general NN architectures have been studied, jNN and dNN where in the former both the outcome and treatment are modeled jointly (with an appropriate loss function) and in the latter, they are modeled separately. We have observed that $L_{1}$ regularization especially the ones that targets the treatment model ($L_{1TG}$) is an effective hyperparameter for achieving a better bias-variance trade-off for the normalized Augmented Inverse Probability Weighting (nAIPW) estimator. And, the number of neurons in the first and last layer of the network becomes irrelevant as long as the value of $L_{1TG}$ is sufficient. Further, we have observed that in the hyperparameter settings where the IV effects are controlled, the estimation is less biased and more stable. Thus the targeted regularization is successful in dampening the IVs and preventing perfect prediction in the treatment model. Figure 3, Figure 4, Figure 5, Figure 6, Figure 7 and Figure 8 illustrate that jNN is overall more stable and has a smaller RMSE in the small sample dataset scenario as compared to dNN. We utilized nAIPW in our simulations as they outperform or at least do not underperform AIPW and other estimators such as IPW, nIPW, AIPW, and SR. The nAIPW estimator has a normalization factor in the denumerator which can dilute the impact of extreme predictions of the propensity score model and protect the estimator against the positivity assumption violation Van der Laan and Rose [18].

We utilized a geometric-type average of the $R^{2}$ and AUC to choose among the first step models. As the objective of optimization in the first step is increasing prediction performance which is not necessarily the same as the causal inference objectives, the usage of either $R^{2}$, AUC or their geometric average is sub-optimal. In a future study alternative approaches will be explored and compared with the said prediction measure.

A real strength of NNs would be to uncover hidden information (and thus confounder effects) in unstructured data such as text or image data. However, in this article, we have not studied the presence of unstructured data and it is left for future research.

There are limitations due to the assumptions and simulation scenarios and, thus, some questions are left to future studies to be explored. For example, the outcome here was assumed to be continuous, and the treatment to be binary. We also did not cover heavy tail outcomes or rare treatment scenarios. Also, the ratio of dimension to the size of the data was considered to be fairly small (p≪n), and we have not studied the case where n<p. Furthermore, we did not study the asymptotic behavior of nAIPW when jNN or dNN predictions are used.

A limitation of jNN as compared to dNN is that if one needs to shrink the final hidden layer to control the complexity of the treatment model, by structure, we are limiting the complexity of the outcome model which might not be necessary. This might be resolved by another architectural design, which is left to future studies on the subject.

The usage of another regularization technique that controls the extremeness of propensity score values is a plausible approach. For example, a data-dependent term can be added to the loss function $\sum_{i=1}^{n}\frac{1}{g_{i}}+\frac{1}{1-g_{i}}.$ Such a term discourages the network to obtain values extremely close to zero or one, as opposed to the negative log-likelihood term that encourages such tendencies. This approach might also focus less on the inputs that cause extreme values such as strong confounders or IVs. Examination of this approach is left to future studies.

In the design of the optimization, we did not consider a formal early stopping as a regularization technique. However, in the preliminary exploration, our simulations performed better with fewer iterations (in fact epochs). In modern NNs, researchers usually run the NN algorithms in many iterations, but that is partly due to the dropout regularization technique. We did not use drop-out (and L2) regularization in the final simulations, as the preliminary results did not confirm dropout as promising as $L_{1}$ regularization.

Further, we utilized NNs to learn the underlying relationships between the covariates and the outcome and treatment by targeting the relevant features through regularization and joint modeling of the treatment and outcome. NNs with other structures that might target confounders have not been explored, nor have other Machine Learning algorithms such as tree-based models. The Gradient Boosting Machines (GBM) algorithm Friedman [41] can be alternatively used to learn these non-linear relationships while targeting the right set of features. This is postponed to a future article.

## Figures and Tables

**Figure 1 entropy-24-01290-f001:**
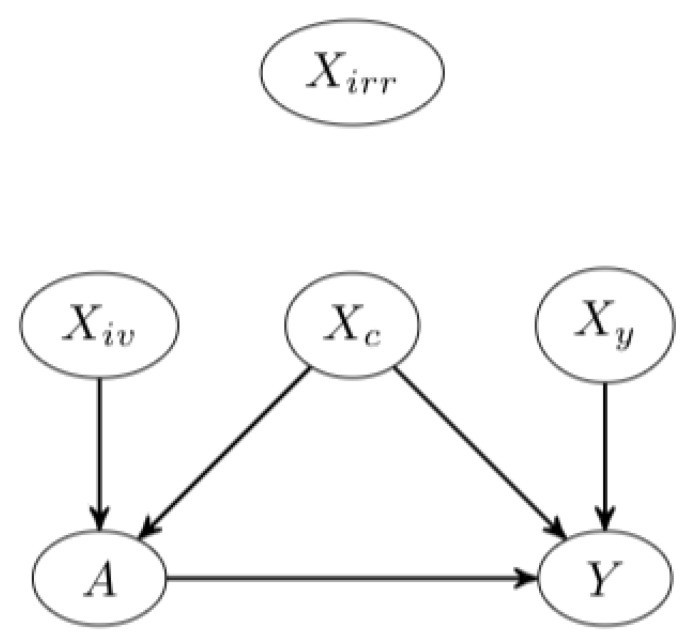
The causal relationship between *A* and *y* in the presence of other factors in an observational setting.

**Figure 2 entropy-24-01290-f002:**
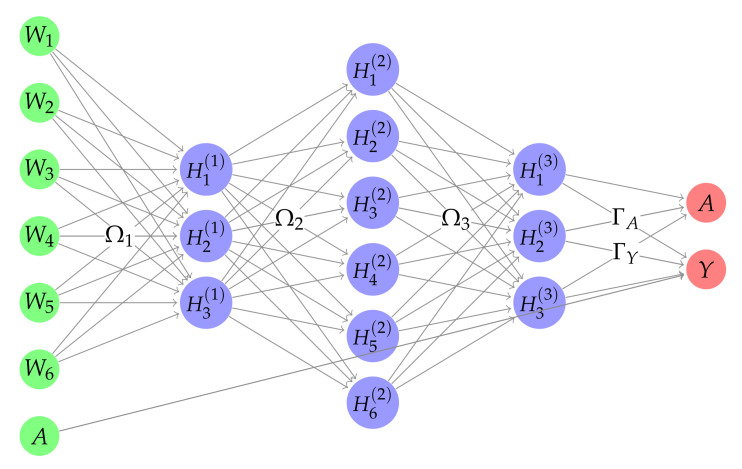
A Joint Neural Network architecture that incorporates linear effect of the treatment on the outcome, and the nonlinear relationship between the covariates and the treatment assignment and the outcome, all three tasks at the same time.

**Figure 3 entropy-24-01290-f003:**
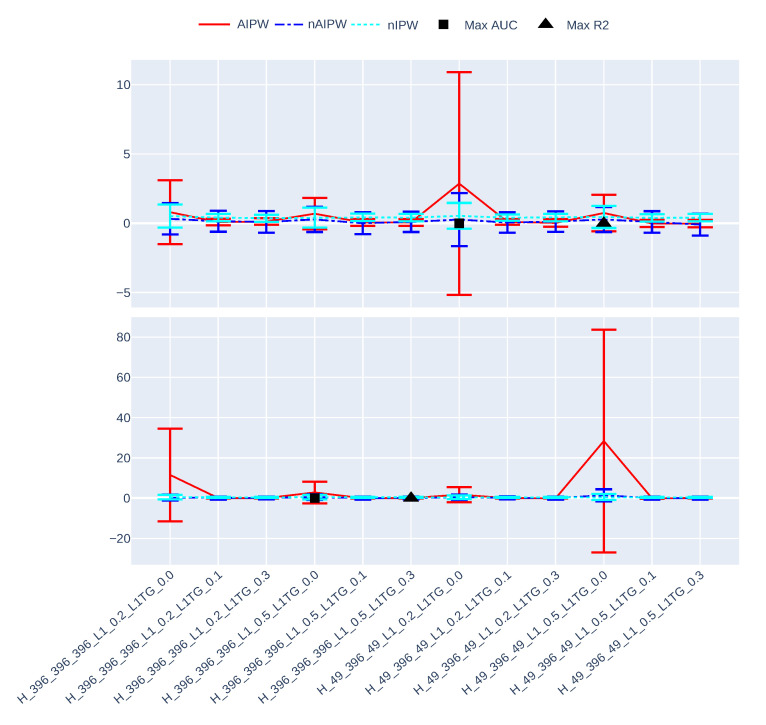
The ATE estimates and their asymptotically calculated 95% confidence intervals with nIPW, AIPW, and nAIPW methods.

**Figure 4 entropy-24-01290-f004:**
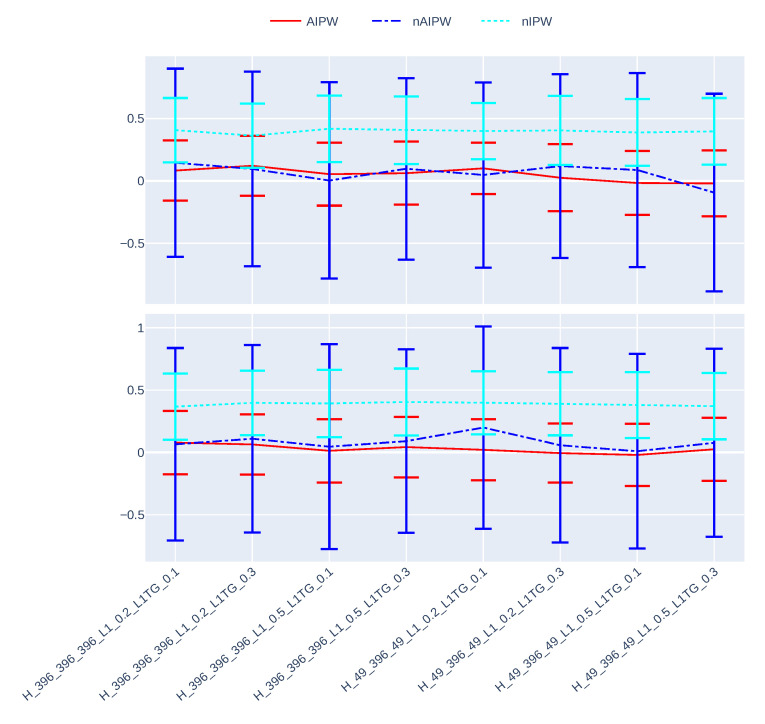
The ATE estimates and their asymptotically calculated 95% confidence intervals with nIPW, AIPW, and nAIPW methods.

**Figure 5 entropy-24-01290-f005:**
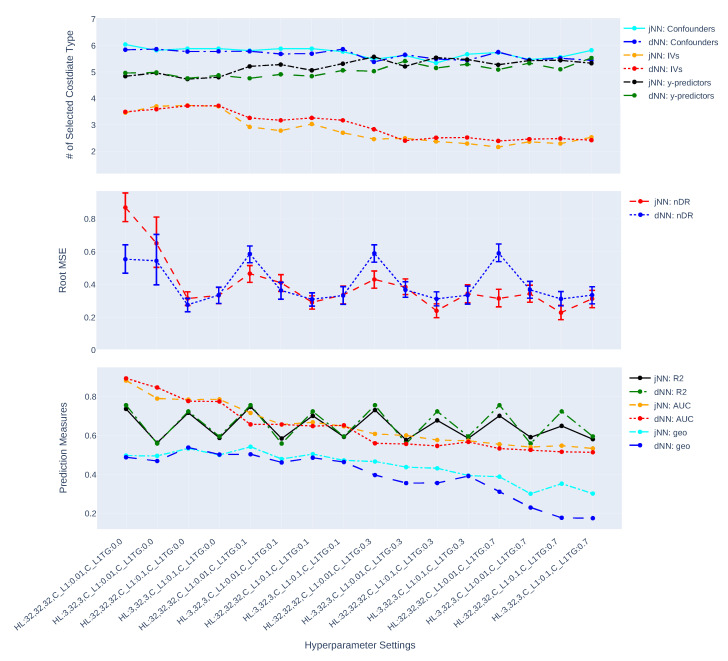
The comparison of captured number of confounders, IVs and y-predictors, RMSE of nAIPW and its bootstrap 95% confidence interval, and prediction measures $R^{2}$, AUC and geo (geometric mean of $R^{2}$, AUC) for different hyperparameter settings and where the predictions come from jNN or dNN models. (n = 750, *p* = 32).

**Figure 6 entropy-24-01290-f006:**
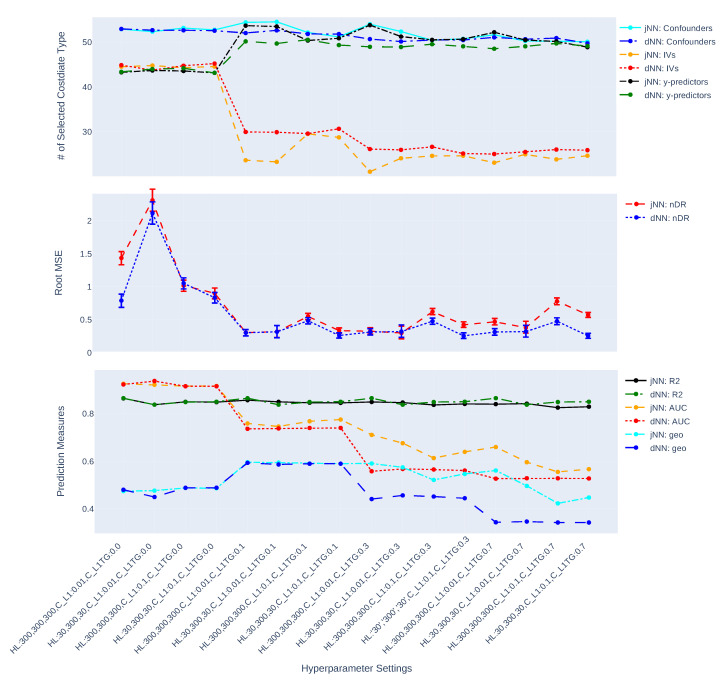
The comparison of captured number of confounders, IVs and y-predictors, RMSE of nAIPW and its bootstrap 95% confidence interval, and prediction measures $R^{2}$, AUC and geo (geometric mean of $R^{2}$, AUC) for different hyperparameter settings and where the predictions come from jNN or dNN models. (n = 7500, *p* = 300).

**Figure 7 entropy-24-01290-f007:**
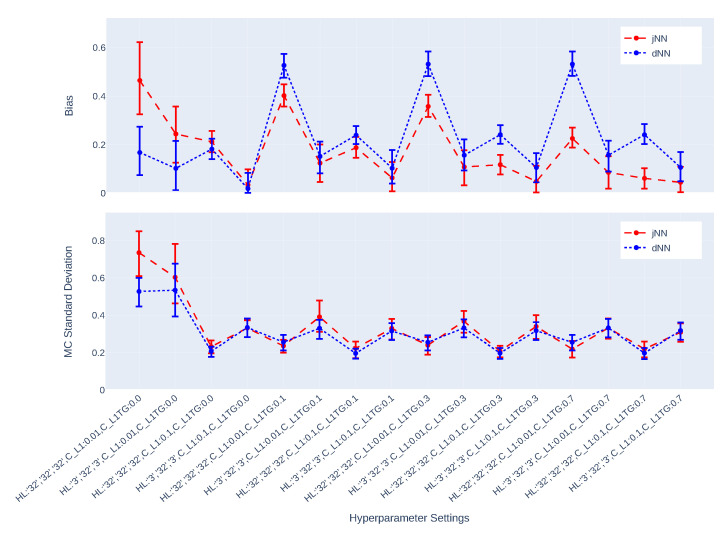
The bias and standard deviation of nAIPW and their bootstrap 95% confidence intervals for different hyperparameter settings where the predictions come from jNN or dNN models. (n = 750, *p* = 32).

**Figure 8 entropy-24-01290-f008:**
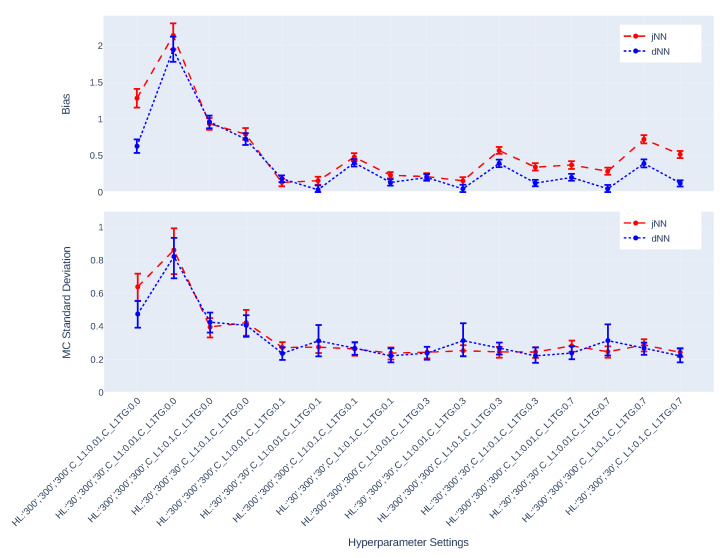
The comparison of bias, Monte Carlo standard deviation and their bootstrap 95% confidence intervals of nAIPW, for different hyperparameter settings and the predictions come from jNN or dNN models. (n = 7500, *p* = 300).

## Data Availability

The simulated data can be regenerated using the codes, which can be provided to the interested user via an email request to the correspondence author. The CCHS data is not publicly available and only the authorized people can access and perform analyses on it.

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
