# Peer review of "Targeted *L*_1_-Regularization and Joint Modeling of Neural Networks for Causal Inference"

_entropy, 2022, doi:10.3390/e24091290_

Round 1

Reviewer 1 Report

(1) The paper should cite a review article on double robust estimation. The best reference would be arXiv:2105.02071.

(2) On page 2 line 16, and the following discussion, the paper says that dNN without regularization does not perform well. This should be changed because the simulations in the cited reference [17] actually show it works well to recover the causal effect. Further, the simulation evidence in this very submission show that dNN works well when correctly implemented.

(3) The paper must cite and discuss arXiv:1309.4686. That paper contains discuss of selecting covariates for efficiency, not conditioning on instruments, and the correct bounds to attain.

(4) The paper should be clear that there is no reason to expect the outcome regression and propensity score to share basis functions, which is what is imposed in Figure 2 and equation 2. Again please cite and refer to arXiv:1309.4686 for discussion of when the outcome function and propensity score share *covariates*, but they would not share basis functions.

(5) The paper must cite and discuss arXiv:1906.02120 because that paper does almost the same idea as this. Without proper discussion it will appear that the current submission is stealing their ideas.

(6) The paper must cite and discuss arXiv:2010.14694. That paper develops more general double/local robust estimators. The methods and simulation results of this submission would be useful in these more general settings. If possible, the paper should conduct simulations in a more general model, beyond simple binary treatments.

(7) It should be pointed out in the paper that some of the references here, such as [17], use early stopping, and this is a form of regularization.

(8) The paper should cite arXiv:2009.06111 when discussing dropout regularization.

(9) The discussion of the results should be more balanced. The evidence is not as strongly in favor of the proposed method as it is described.

Author Response

Thank you for your time and your comments. We have addressed the comments and necessary changes below in the subsequent pages and the revised manuscript.

However, we have not added new material to our findings as they are not only beyond the scope of this research but there is not enough time to add that material within the short time we have to revise the changes. However, we have added sufficient explanations throughout the manuscript to clarify the objectives and findings.

Reviewer 2 Report

Major comments:

- there is a ton of work on using deep learning for inference of the ATE, with and without debiasing strategies like AIPW (https://scholar.google.com/scholar?q=neural+network+semiparametric+efficient+average+treatment+effect&hl=en&as_sdt=0&as_vis=1&oi=scholart). Not all of this work is high-quality, but a broader textual summary would be appreciated in the introduction. Of particular relevance are Shi 2019 (https://arxiv.org/abs/1906.02120) and  Chernozhukov 2022 (https://proceedings.mlr.press/v162/chernozhukov22a.html). Specifically, Shi 2019 proposes the same kind of multitask architechture that is proposed in this paper.

- the experiments only compare the dNN and jNN methods across different hyperparameter settings. It would be helpful for the reader to be able to contextualize those results by comparing them to AIPW using off-the-shelf methods (e.g. random forests). More importantly, when using AIPW it is standard to use a (cross-estimated) cross-validated ensemble over a diversity of learners (sometimes called "super learner; see eg https://arxiv.org/pdf/2004.10337.pdf). The authors should thus compare the performance of these individual methods to the performance of a method that uses CV to select between each of these learners for each model separately. This could highlight the potential benefit of the jNN-type approach, which is that the model selection doesn't happen independently for propensity score and outcome model. This is also related to the cTMLE method (https://www.ncbi.nlm.nih.gov/pmc/articles/PMC2898626/). This would also clarify how much of the purported benefit comes from this joint modeling vs from the L1 regularization vs from a synergistic combination of the two.

- the figures are difficult to read and interpret in part because all of the hyperparameter settings are jammed into labels on the x-axis. I recommend using facets or color for the hyperparameters and reducing the number of metrics considered in the main paper (e.g. just look at prediction RMSE insead of R2 and AUC, etc.). The figures should be targeted to emphasize the main takeaways rather than trying to summarize every single simulation that was run. The other metrics/figures can go in a supplement.

Minor comments:

- CATE and ATE are not the only two causal estimands as the introduction implies. For example, most IV analyses target the complier average effect, not the population ATE. Other estimands include the local ATE (estimated in most reg. discontinuity analyses), average outcome under a particular treatment policy, and average outcome under the optimal policy. The salient distinction is not between ATE and CATE, but between properties of the true causal distribution that can be estimated at root-n rates (which tend to be averages) and those that cannot (which tend to be functions). This comes down to pathwise differentability. For more see, eg https://arxiv.org/abs/2203.06469

- efficient estimation of the ATE (and other parameters) has a long history that should be credited in the introduction. A good summary of this history can be found here: https://vanderlaan-lab.org/2019/12/24/cv-tmle-and-double-machine-learning/

- around line 120 the standard identifying assumptions are given for ATE. However, these are not the only possible identifying assumptions (i.e. it is not correct to say that these must be satisfied). For example, under some parametric assumptions and with an IV, the ATE is identified even if there is unobserved confounding. Not a big deal, but should be clarified.

- the idea of regularizing out IVs has previously been done "manually" by first selecting  variables that are correlated with the outcome and then using just those in the propensity model (see eg Schneeweiss "high dimensional propensity score"). This history should be noted.

Author Response

Dear reviewer

Thank you for your time and report. We have addressed the comments and necessary changes below in the subsequent pages and the revised manuscript.

However, we have not added new material to our findings as they are not only beyond the scope of this research but there is not enough time to add that material within the short time we have to revise the changes. However, we have added sufficient explanations throughout the manuscript to clarify the objectives and findings.

Regards

Round 2

Reviewer 2 Report

all comments are addressed

Author Response

Dear Reviewer

Thanks for the comments regarding the spell-checking. I've run a spell checker and hopefully have captured all the typos.

Best regards
